# The Use of near Infra-Red Radiation Imaging after Injection of Indocyanine Green (NIR–ICG) during Laparoscopic Treatment of Benign Gynecologic Conditions: Towards Minimalized Surgery. A Systematic Review of Literature

**DOI:** 10.3390/medicina58060792

**Published:** 2022-06-13

**Authors:** Antonio Raffone, Diego Raimondo, Alessia Oliviero, Arianna Raspollini, Antonio Travaglino, Marco Torella, Gaetano Riemma, Marco La Verde, Pasquale De Franciscis, Paolo Casadio, Renato Seracchioli, Antonio Mollo

**Affiliations:** 1Department of Medical and Surgical Sciences (DIMEC), Univertity of Bologna, 40138 Bologna, Italy; anton.raffone@gmail.com (A.R.); arianna.raspollini@gmail.com (A.R.); renato.seracchioli@aosp.bo.it (R.S.); 2Division of Gynaecology and Human Reproduction Physiopathology, IRCCS Azienda Ospedaliera-Universitaria di Bologna, 40138 Bologna, Italy; 3Gynecology and Obstetrics Unit, Department of Medicine, Surgery and Dentistry Schola Medica Salernitana, University of Salerno, 84081 Baronissi, Italy; alessia.oliviero@hotmail.com (A.O.); antmollo66@gmail.com (A.M.); 4Unità di Ginecopatologia e Patologia Mammaria, Dipartimento Scienze della Salute della Donna, del Bambino e di Sanità Pubblica, Fondazione Policlinico Universitario A. Gemelli IRCCS, 00168 Rome, Italy; antonio.travaglino.ap@gmail.com; 5Obstetrics and Gynecology Unit, Department of Woman, Child and General and Specialized Surgery, University of Campania Luigi Vanvitelli, 80128 Naples, Italy; marcotorella@iol.it (M.T.); gaetano.riemma7@gmail.com (G.R.); marco.laverde88@gmail.com (M.L.V.); pasquale.defranciscis@unicampania.it (P.D.F.)

**Keywords:** laparoscopy, innovation, technology, gynecology, minimally-invasive, fluorescent dye, fluorescence

## Abstract

*Background and Objectives*: To assess the use of near infrared radiation imaging after injection of indocyanine green (NIR–ICG) during laparoscopic treatment of benign gynecologic conditions. *Materials and Methods*: A systematic review of the literature was performed searching 7 electronic databases from their inception to March 2022 for all studies which assessed the use of NIR–ICG during laparoscopic treatment of benign gynecological conditions. *Results*: 16 studies (1 randomized within subject clinical trial and 15 observational studies) with 416 women were included. Thirteen studies assessed patients with endometriosis, and 3 studies assessed non-endometriosis patients. In endometriosis patients, NIR–ICG use appeared to be a safe tool for improving the visualization of endometriotic lesions and ureters, the surgical decision-making process with the assessment of ureteral perfusion after conservative surgery and the intraoperative assessment of bowel perfusion during recto-sigmoid endometriosis nodule surgery. In non-endometriosis patients, NIR–ICG use appeared to be a safe tool for evaluating vascular perfusion of the vaginal cuff during total laparoscopic hysterectomy (TLH) and robotic-assisted total laparoscopic hysterectomy (RATLH), and intraoperative assessment of ovarian perfusion in adnexal torsion. *Conclusions*: NIR–ICG appeared to be a useful tool for enhancing laparoscopic treatment of some benign gynecologic conditions and for moving from minimally invasive surgery to minimalized surgery. In particular, it might improve treatment of endometriosis (with particular regard to deep infiltrating endometriosis), benign diseases requiring TLH and RATLH and adnexal torsion. However, although preliminary findings appear promising, further investigation with well-designed larger studies is needed.

## 1. Introduction

Indocyanine green (ICG) is a fluorescent dye which binds plasma proteins in the vascular system [1,2,3]. Kodak laboratories invented ICG dye for near infra-red (NIR) photography in 1955 and it was later approved by the FDA for clinical use in 1959 [4]. Once in the blood flow, ICG rapidly bounds to lipoproteins and it is almost entirely extracted by the liver appearing visible in the bile 8 min after the injection. When ICG in not injected in the blood stream, it reaches the nearest draining lymph node in approximately 15 min [5,6]. ICG use is safe, with a dose of 0.1–0.5 mg/mL/kg for clinical use [4]. Thanks to its ability to assess tissue vascularization once detected with a specific wavelength of light, NIR imaging with ICG injection (NIR–ICG) has proven a useful, feasible and safe tool during gynecologic, urologic and digestive procedures for both benign and malignant diseases [1,2,7,8]. In particular, NIR–ICG can be used for identifying sentinel lymph nodes during surgical staging for several cancers (melanoma, prostate, rectal or endometrial cancer) [9,10,11].

On the other hand, for benign conditions, NIR–ICG can be used with several applications. For example, it can be used as a guide during endometriosis surgery facilitating intraoperative diagnosis of occult peritoneal and deep endometriotic lesions at white light [7,12,13]. Furthermore, it has been proven useful in the evaluation for anastomotic perfusion assessment after discoid or segmental resection for rectosigmoid endometriosis (RSE) [14,15]. NIR–ICG dye may also help in the intraoperative assessment of organ perfusion and ischemia after ovarian detorsion and assist the surgeon’s intraoperative decision [16]. Despite the several proposed applications for NIR–ICG in benign gynecologic conditions, to our knowledge, systematic assessment of evidence in this field is lacking in the literature. The aim of this study is to assess the use of NIR–ICG during laparoscopic treatment of benign gynecologic conditions through a systematic review of the literature.

## 2. Materials and Methods

### 2.1. Study Protocol

The study followed an a priori designed study protocol, with each review step independently completed by 2 authors. Discussion with senior authors was used as a method to solve disagreements. 

The Preferred Reporting Item for Systematic Reviews and Meta-analyses (PRISMA) statement and checklist was used for reporting the whole study (see Appendix A) [17].

### 2.2. Search Strategy and Study Selection

MEDLINE, Web of Sciences, Google Scholar, Scopus, ClinicalTrial.gov, Cochrane Library, and EMBASE were searched as electronic databases from their inception to March 2022 using different combinations of the following text words: “indocyanine green’’, “ICG’’, “NIR–ICG”; “near infrared”; ‘’fluorescence’’, “firefly’’, “arter’’, “angiographic’’, “vascular’’, “ischem*’’, “anastomo*’’, “perfusion’’, “laparoscop*’’, “gynecol*’’, “gynaecol*’’, “myom*’’, “fibrom*’’, “uter’*’, “ovar*’’, “endometr*’’, “adenomyo*’’. An example of search strategy (adopted for the MEDLINE) was the following: (indocyanine green OR ICG OR NIR–ICG OR near infrared OR fluorescence OR firefly) AND (arter* OR angiographic OR vascular* OR ischem* OR anastomo* OR perfusion) AND laparoscop* AND (gynecol* OR gynaecol* OR myom* OR fibrom* OR uter* OR ovar* OR endometr* OR adenomyo*).

References from each full-text assessed study were also screened for missed studies. All peer-reviewed studies which assessed the use of NIR–ICG during laparoscopic treatment of benign gynecological conditions were included. We a priori excluded:−case reports;−literature reviews;−studies in languages other than English;−video articles;−studies which assessed the use of NIR–ICG in gynecologic malignancies or in non-gynecologic conditions.

### 2.3. Risk of Bias within Studies Assessment

Two authors independently assessed the risk of bias within the included studies via the Methodological Index for Non-Randomized Studies (MINORS). In detail, the following seven applicable domains were considered for the risk of bias in each study: (1) A clearly stated aim (i.e., if the question addressed is precise and relevant); (2) Inclusion of consecutive patients (i.e., if all patients potentially fit for inclusion were included in the study during the study period); (3) Prospective collection of data (i.e., if data were collected according to a protocol established before the beginning of the study); (4) Endpoints appropriate to the aim of the study (i.e., if explanation of the criteria used to evaluate the outcomes was unambiguous); (5) Unbiased assessment of the study endpoints (i.e., if the assessment of study endpoints was unbiased); (6) Follow-up period appropriate to the aim of the study (i.e., if the follow-up was sufficiently long to allow the assessment of the endpoints); (7) Loss to follow up less than 5% (i.e., if patients lost to follow up were less than 5% of total population.

Authors judged each domain for each included studies as “low risk”, “high risk” or “unclear risk” of bias based on data were “reported and adequate”, “reported but inadequate” or “not reported”, respectively.

### 2.4. Data Extraction

Data from included study were extracted without modification according to the PICO (Population, Intervention or risk factor, Comparator, Outcomes) items.

“Population” of our study was women with benign gynecological conditions.

“Intervention” was the use of NIR–ICG during laparoscopic treatment.

“Comparator” was the non-use of NIR–ICG during laparoscopic treatment.

“Outcome” was the improvement in surgical laparoscopic outcomes.

Review Manager 5.3 (Copenhagen: The Nordic Cochrane Centre, Cochrane Collaboration, 2014) was used as a software [18].

## 3. Results

### 3.1. Study Selection

After electronic database searches, 345 articles were identified. Sixty articles remained after duplicate removal, 45 after title screening and 38 after abstract screening; these were evaluated for eligibility. Twenty-two articles were then excluded based on the above-reported a priori exclusion criteria. Finally, 16 articles with 416 women were included in our study (Figure 1). 

### 3.2. Studies and Patients’ Characteristics

Of the included studies, one study [19] was a prospective, single-center, randomized within subject clinical trial, while 15 studies were observational: 4 retrospective [14,15,20,21] and 11 prospective [1,2,7,12,16,20,22,23,24,25,26,27]. 

Thirteen studies assessed patients with endometriosis [1,2,7,12,14,15,19,20,21,22,23,24,25], with the following NIR–ICG applications:−to localize ureteral course (2 studies [20,23]);−to assess ureteral perfusion after conservative surgery (1 study [1]);−to improve visualization of endometriotic lesions (6 studies [7,12,19,21,22,24]);−to evaluate the different rectosigmoid endometriosis (RSE) vascular patterns and the correlation with clinicopathological data (1 study [2]);−to assess bowel vascularization after deep infiltrating endometriosis (DIE) surgery to reduce the risk of fistula (2 studies after full-thickness bowel resection [14,15] and 1 study after shaving technique [25]).

The remaining 3 studies [16,26,27] assessed non-endometriosis patients; in particular::−2 studies assessed the NIR–ICG capacity to visualize the vascular perfusion of the vaginal cuff after total hysterectomy in order to decrease vaginal cuff dehiscence rate [26,27].−1 study assessed if the NIR–ICG was a faceable tool to evaluate intraoperatively ovarian perfusion after detorsion [16] (Table 1).

Details about benign gynecologic conditions were reported in Table 1.

Regarding the study population, the mean age ranged from 25 to 36 years, while mean BMI ranged from 22.8 to 35.4 kg/m^2^; 14.9% of patients had at least one child. The mean intraoperative time ranged from 121 to 163.5 min, with a median estimate of blood loss which ranged from 50 to 150 mL. The follow-up time ranged from 1 to 23 months. Indication for surgery was dysmenorrhea in 43.8% of patients, dyspareunia in 34.5%, dyschezia 27.6%, menorrhagia in 26.7%, rectorrhagia in 3.1%, ovarian cysts in 30%, infertility in 6.8%, pelvic pain and infertility in 20.63%, renal colic in 6.6%, hydroureter in 20%, hydroureteronephrosis in 13.3%, abnormal uterine bleeding (AUB) in 42.5%, cervical dysplasia in 15%, Lynch syndrome in 5%, postmenopausal bleeding in 5% and pelvic pain in 39.3% (Table 2).

Laparoscopy was robot-assisted in 5 studies [2,12,20,26,27]. Indocyanine injection was intraurethral in 2 studies [20,23], intravenous in 13 studies and both intraurethral and intravenous in one study [24]. Time to NIR–ICG visualization ranged from 6 to 9 min for intraurethral injection and from 5 s to 30 min for intravenous injection. No complication due to NIR–ICG injection was reported in the included studies.

Details about indocyanine dosage range and type of surgery were reported in Table 3.

### 3.3. NIR–ICG Performance

In endometriosis patients, NIR–ICG use appeared to be a safe tool for:−improving visualization of endometriotic lesions and ureters, preventing iatrogenic injuries after its intraurethral injection [20,23];−supporting surgeons in surgical decision-making process with the assessment of ureteral perfusion after conservative surgery [1];−improving endometriosis identification, with particular help in (1) separating the healthy rectal tissue from the rectovaginal DIE nodules (RVDIEN) [22], (2) decision whether to enlarge the resection to the posterior vaginal fornix in case of RVDIEN [22], (3) in the resection of deep infiltrating nodules [22]; such improvement was not found in one study [19];−intraoperatively assessing bowel perfusion during recto-sigmoid endometriosis nodules (RSE) surgery, with improvement in patient safety, intraoperative decision-making process and surgical outcomes [2,14,15,25].

In non-endometriosis patients, NIR–ICG use appeared to be a safe tool for:−evaluating vascular perfusion of the vaginal cuff during total laparoscopic hysterectomy (TLH) and robotic-assisted total laparoscopic hysterectomy (RATLH), with help in understanding causes for vaginal cuff dehiscence; however, an improving in methods for quantification of fluorescence might be needed to utilize it for clinical use [26,27];−intraoperative assessment of ovarian perfusion in adnexal torsion [16].

### 3.4. Risk of Bias within Studies Assessment

For the “A clearly stated aim” and “Follow-up period appropriate to the aim of the study” domains, all the included studies were categorized at low risk of bias.

For the domain “Inclusion of consecutive patients”, seven studies did not report if all eligible patients were included in the study during the study period therefore, they were classified at unclear risk of bias [7,12,20,21,22,26,27]. The other studies were at low risk of bias.

Regarding the “Prospective collection of data”, all studies were considered at low risk of bias except for one study that was at unclear risk of bias because it was not clear if the data were collected according to a protocol established before the beginning of the study [20].

For the “Endpoints appropriate to the aim of the study” and “Unbiased assessment of the study endpoint” domains, 2 studies were considered at unclear risk of bias because they did not clearly state the study outcomes and it was unclear if the assessment of study endpoints was unbiased [20,22].

For “Loss to follow up less than 5%” two studies were considered at unclear risk because it was not clearly stated if all the patients completed their follow up period [1,7]; four studies were evaluated at high risk of bias because more than 5% of the patients were lost during the follow up period [16,23,26,27].

Results about risk of bias within study assessment were graphically shown in Figure 2.

## 4. Discussion

### 4.1. Main Findings

This study showed that NIR–ICG might be a safe tool for improving laparoscopic treatment of some benign gynecologic conditions. In particular, it might enhance surgery for endometriosis women, with improvement in: visualization of endometriotic lesions and ureters, surgical decision-making process and assessment of bowel perfusion. Such improvements seem to benefit even more complex laparoscopic surgery for DIE. Regarding other benign gynecologic conditions, NIR–ICG appeared to support TLH and RATLH providing the chance of evaluating vascular perfusion of the vaginal cuff and laparoscopic treatment of adnexal torsion with the assessment of ovarian perfusion.

### 4.2. NIR_ICG History

Over the last few years, the clinical role of NIR–ICG has clearly increased also due to its capacity to visualize tissue and organ perfusion in real-time. Moreover, it has been proven that it is a nontoxic substance with a short lifetime, allowing for repeated administrations [4].

The first applications of this technique were in the measurement of liver function, the study of cardiac output and in the detection of choroidal vascularization; later, it has been used to estimate vascularization of colorectal anastomoses [28].

### 4.3. NIR–ICG Application in Gynecological Conditions

Still later, NIR–ICG has been widely studied and employed in laparoscopic treatment of benign gynecological conditions [1,2,7,12,14,15,16,19,20,21,22,23,24,25,26,27].

#### 4.3.1. Endometriosis

In particular, the main field of application has been endometriosis, with specific regard to DIE [1,2,7,20,21,22,23,24,25]. In fact, DIE surgery is challenging and can be associated with major and minor complications, such as hemorrhage, infections, nerve damage, laparotomic conversion, fistula and bladder and bowel dysfunction [29,30,31,32,33,34,35,36,37]. Therefore, enhancing such surgery with innovative tools able to reduce complications rate appears to be a priority. In detail, in DIE surgical treatment, NIR–ICG has been assessed with several applications. 

##### Localization of Ureteral Course

First, it has been assessed as a tool to localize ureteral course and to prevent iatrogenic injuries during complex laparoscopic surgery [20,23]. In fact, iatrogenic intraoperative ureteral injury is one of the most common avoidable complications of laparoscopic gynecological surgery, with an incidence of 7.6% [23,37]. When compared to methods for intraoperative ureteral identification (i.e., conventional DJ ureteral stents or illuminated ureteral catheters), NIR–ICG shows the advantage of avoiding a complete ureteral catheterization with related complications. Furthermore, it appears cheap and easy to be performed even in the absence of a urologist [23]. However, given the small sample size of the studies assessing NIR–ICG for this application [20,23], further studies are necessary for validating this promising role in supporting endometriosis surgery.

Regarding NIR–ICG application in ureteral assessment, it has been also studied for evaluating ureteral vascularization during endometriotic surgery, concluding that it can be a helpful tool for preventing any useless stent positioning and its related complications after ureterolysis for DIE [1].

##### Endometriosis Identification and DIE

Later, NIR–ICG has been evaluated as a tool for improving identification of endometriosis based on the known neovascularization of endometriosis lesions [7,12,20,21,22,23,24,25].

In particular, Cosentino et al. reported that it may be used for an intraoperative endometriosis diagnosis, both confirming visible endometriosis lesions and identifying occult endometriosis lesions that white light evaluation had misinterpreted [7]. In this study, it showed sensitivity of 82.0%, specificity of 97.9%, positive predictive value of 97.8% and negative predictive value of 82.3% in identification of endometriosis lesions; sensitivity even increased to 89% considering DIE alone [7]. However, the fact that 20 pathologic lesions (20.1%) were not confirmed intraoperatively with NIR–ICG implied that it cannot totally replace the white light evaluation but could be used together with it to detect occult lesions. In fact, removing occult disease leads to a decrease inpostoperative pain and a risk of persistence and/or relapse of symptoms [7]. 

The use of NIR–ICG in addition to white light evaluation was also supported by Lier et al. [19] and Vizzielli et al. [21]. Additionally, Jayakumara et al. conclude that NIR–ICG could help surgeons to better visualize, diagnose and treat endometriosis [11]. On the other hand, De Neef et al. found that NIR–ICG may be helpful in achieving a macroscopic resection of RVDIEN, allowing the operators to differentiate peritoneal endometriosis from healthy rectal tissue and thus reduce the risk of rectal perforation [22]. However, additional data are necessary to confirm these preliminary promising results.

Conversely, Siegenthaler et al. described that even though NIR–ICG may be helpful in the resection of deep infiltrating nodules by providing better demarcation from the surrounding healthy tissue, its diagnostic value in detecting and confirming occult endometriosis is minimal, with a reported sensitivity of 14.7% [24]. These contrasting findings might be explained by a different prevalence of some parameters negatively impacting upon the endometriosis detection rate with NIR–ICG, such as a lower ICG exposure time, a higher number of previous abdominal surgery, more advanced-stage endometriosis and a prolonged adhesiolysis [24]. 

Such a low diagnostic value in detecting and confirming occult endometriosis might regard even more avascular or hypovascular pattern nodules [2]. However, these nodules might be identified thanks to the contrast with the surrounding more vascularized tissue [12]. In any case, as many mechanisms are involved in the vascularization of endometriosis nodules, such as angiogenesis, inosculation of preformed microvascular network and vasculogenesis, ICG might use of several of them to improve endometriosis nodules identification [38,39,40]. Another NIR–ICG application in endometriosis field has been the assessment of bowel vascularization after DIE surgery to reduce the risk of bowel fistula. In particular, Bourdel et al. assessed this role for NIR–ICG during laparoscopic rectal shaving [25], while Raimondo et al. assessed it through qualitative and quantitative analyses after full-thickness bowel resection for RSE [15]. In fact, bowel fistula shows a pathogenesis related to vascular impairment, and, although difficult, intraoperative estimation of the rectal residual vascularization appears to be a helpful indicator of the risk of postoperative fistula [25,41]. NIR–ICG appears accurate in assessing residual bowel vascularization, with a complementary role to the other available methods to assess bowel after RSE surgery. Indeed, the gas and Blue test methods are effective tools to detect microperforation and real perforation, but are not adequate for vascularization [25]. However, a delayed perfusion or a low blood flow rate is not always easily identifiable even through NIR–ICG assessment.

In the near future, a detailed analysis of perfusion time and intensity (i.e., a quantitative NIR–ICG evaluation) could allow one to overcome this limit. Further studies are necessary indeed, after the promising findings about quantitative NIR–ICG analysis by Raimondo et al. [14].

#### 4.3.2. Non-Endometriosis Conditions

##### Evaluation of the Vascular Perfusion of Vaginal Cuff

Regarding benign gynecologic conditions other than endometriosis, NIR–ICG has been used to evaluate the vascular perfusion of vaginal cuff after RATLH or TLH [26,27]. However, although Beran et al. provided a foundation for ICG dose and measurable outcomes for this application, its clinical utility seems uncertain, with a need for developing improved methods for quantification of fluorescence. In the future, assessment of vascular perfusion of vaginal cuff through NIR–ICG might reduce the incidence of vaginal cuff dehiscence [26,27].

##### Evaluation of Ovarian Perfusion after Adnexal Detorsion

Lastly, in 2022, Nicholson et al. tried to determine the possible use of NIR–ICG in patient with adnexal torsion. In particular, it was used to evaluate tissue viability after detorsion during surgery [16]. Such application tried to overcome limitations related to the current visual assessment of ovarian blood perfusion after detorsion. In fact, the current visual assessment (i.e., tissue color) may not adequately reflect the tissue viability and the real blood supply [16]. However, although NIR–ICG has proven to be safe and inexpensive also for this application, more studies are needed to draw conclusions about the utility and the clinical use of NIR–ICG in this setting [16].

### 4.4. Strengths and Limitations

To our knowledge, this study may be the first systematic review to assess the use of NIR–ICG during laparoscopic treatment of benign gynecologic conditions in the Literature. In fact, previous studies assessed NIR–ICG role only in endometriosis field [28] or in non-gynecologic diseases [42]. As a limitation, our findings may be affected by a low overall quality of the evidence as shown by the risk of bias within studies assessment. Therefore, although promising, NIR–ICG use in gynecologic conditions requires further investigation by future well-designed larger studies. Furthermore, additional and more comparable studies are necessary to perform comparisons and to provide pooled data.

## 5. Conclusions

NIR–ICG appeared to be a useful tool for enhancing laparoscopic treatment of some benign gynecologic conditions and for moving from minimally invasive surgery to minimalized surgery. In particular, it might enhance endometriosis surgery by improving visualization of endometriotic lesions and ureters, the surgical decision-making process, and the assessment of bowel perfusion, with major impact on complex surgery for DIE. Furthermore, NIR–ICG might also help surgeons in evaluating vascular perfusion of the vaginal cuff after TLH and RATLH and ovarian perfusion after laparoscopic treatment of adnexal torsion. However, although promising, NIR–ICG’s role in gynecologic conditions requires further investigation by future well-designed larger studies.

## Figures and Tables

**Figure 1 medicina-58-00792-f001:**
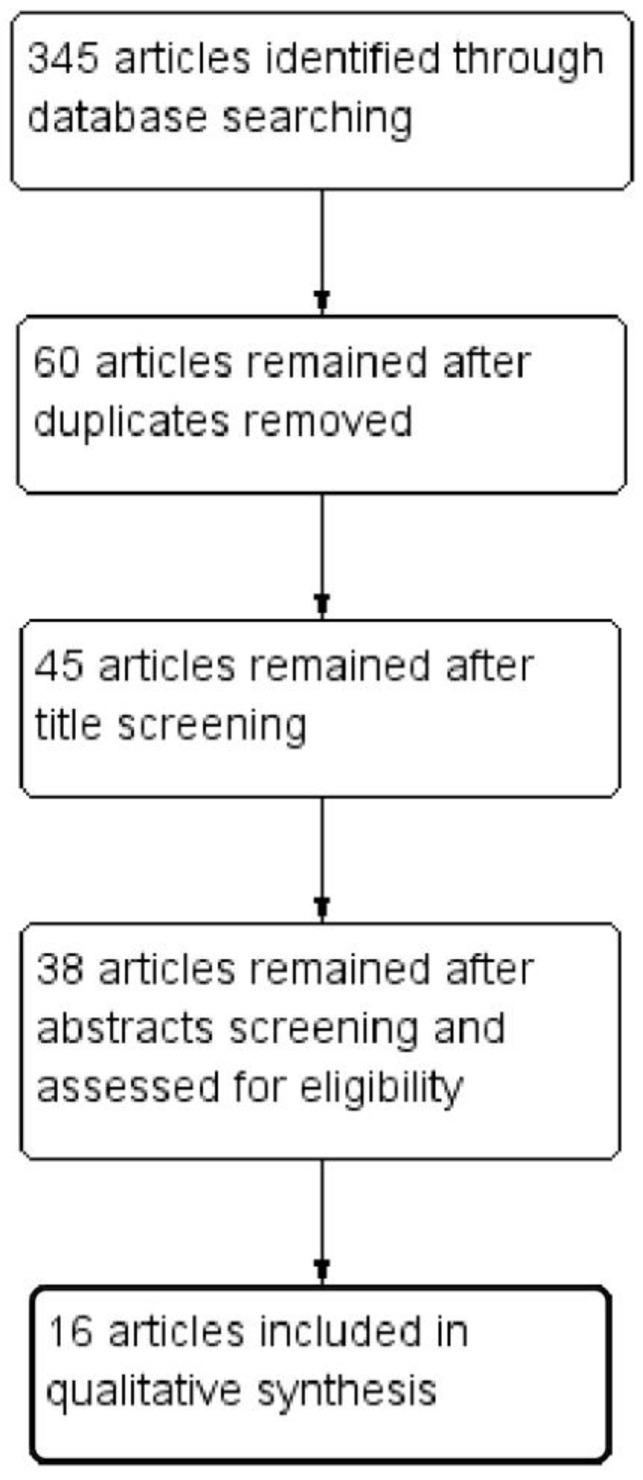
Flow diagram of studies identified in the systematic review (Prisma template [Preferred Reporting Item for Systematic Reviews and Meta-analyses]).

**Figure 2 medicina-58-00792-f002:**
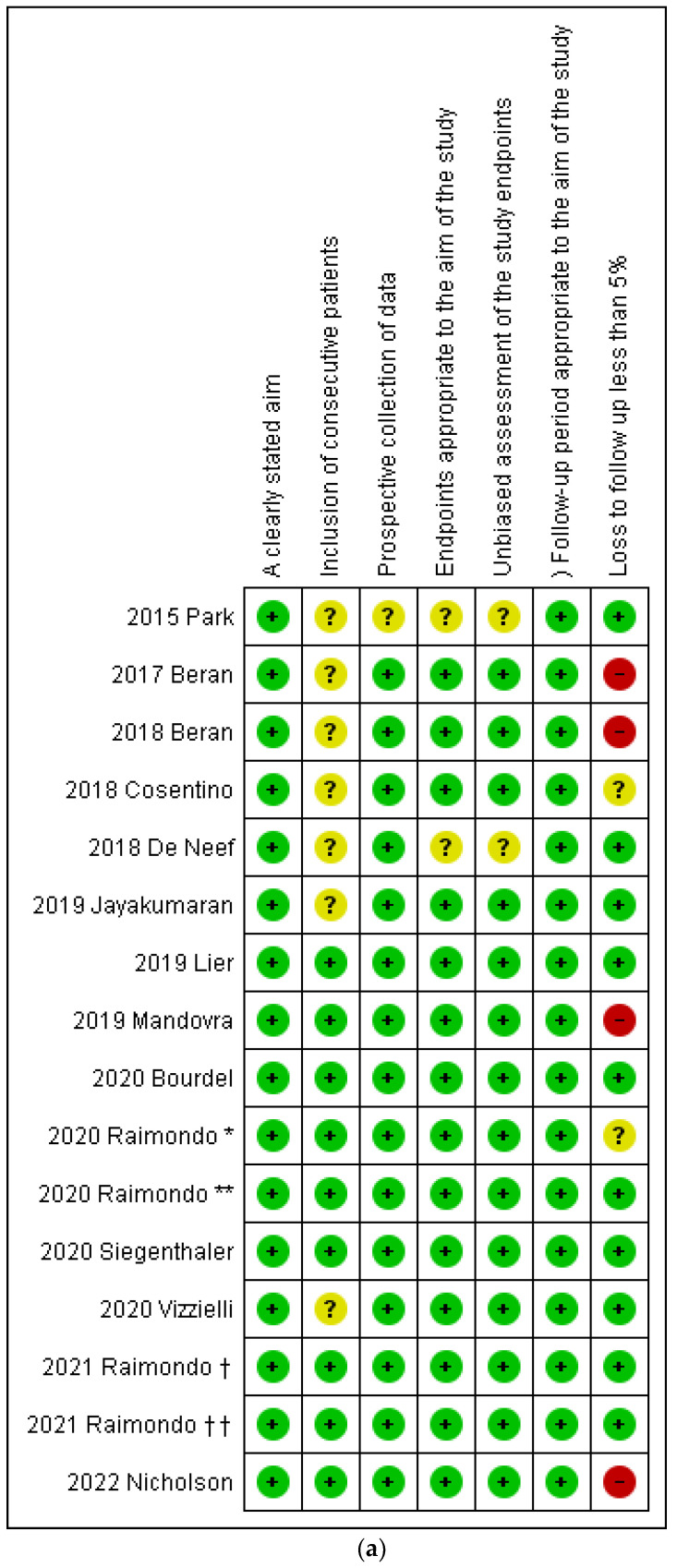
(**a**) Assessment of risk of bias. Summary of risk of bias for each study; plus sign: low risk of bias; minus sign: high risk of bias; question mark: unclear risk of bias. *****: reference no. 1; ******: reference no. 2; **†**: reference no. 14; **††**: reference no. 15. (**b**) Risk of bias graph about each risk of bias item presented as percentages across all included studies.

**Table 1 medicina-58-00792-t001:** Characteristics of the included studies.

Field	ICG Application	Study	Country	Study Design	Sample Size	Study Period	Benign Gynecologic Condition Details	Study Outcomes
**Endometriosis**	To localize ureteral course	2015 Park [20]	USA	Retrospective, observational, cohort study	10	1 July 2014–30 March 2015	DIE	Iatrogenic ureteral injury ICG complications Operative time Estimated blood loss Length of hospital stay
		2019 Mandovra [23]	India	Prospective, observational, cohort study	30	September 2017–December 2017	DIE	Identification of ureters ICG complications Operative time ICG injection time
	To assess ureteral perfusion after conservative surgery	2020 Raimondo [1]	Italy	Prospective, observational case series	36	May 2018–January 2019	DIE (Ureteral)	Ureteral perfusion grade NIR–ICG assessment time Inter-operator agreement regarding ureteral perfusion grade Changes to the surgical plan after NIR–ICG evaluation Perioperative complications Clinical-radiologic outcomes at early follow-up
	To improve endometriosis identification	2018 Cosentino [7]	Italy	Prospective, observational, cohort, single center, single-arm, pilot study	27	January 2016–February 2017	PE-DIE rARSM score -Stage I: 0 -Stage II: 3 -Stage III: 10 -Stage IV: 14	Identified endometriosis lesions
		2018 De Neef [22]	Belgium	Prospective, observational case series	6	-	RVDIE	Resection of RVDIEN Rectal perforations
		2019 Jayakumaran [12]	USA	Prospective, observational cohort study	7	July 2013–June 201	DIE	Identified endometriosis lesions Quality of life
		2019 Lier [19]	The Netherlands	Prospective, single-center, randomized within subject clinical trial	20	February 2016–May 2017	ASRM stage III–IV endometriosis	Detection of peritoneal endometriotic lesions
		2020 Siegenthaler [24]	Switzerland	Prospective, observational, cohort, single-center, single-arm pilot study	63	April 2017–December 2018	PE-DIE No endometriosis: 9 (14.3) rARSM stage (%) -Stage I: 12 (19.0) -Stage II: 10 (15.9) -Stage III: 11 (17.5) -Stage IV: 20 (31.7)	Identified endometriosis lesions
		2020 Vizzielli [21]	Italy	Retrospective, observational, multicenter case-control study	20 cases vs. 27 controls	January 2016–March 2018	PE-DIE -Stage I: 0 -Stage II: 6 (13) -Stage III: 16 (34) -Stage IV: 25 (47)	Visual detection rate of endometriotic lesions
	To evaluate the different RSE vascular patterns and the correlation with clinicopathological data	2020 Raimondo [2]	Italy	Prospective, observational cohort pilot study	30	June 2019–September 2019	DIE (RSE)	Perfusion grade of RSE Preoperative, intraoperative and pathological data
	To assess bowel vascularization after surgery to reduce the risk of fistula	2020 Bourdel [25]	France	Prospective, observational, cohort, single-center, study	23	August 2017–October 2018	Shaving technique for DIE infiltrating the rectovaginal septum	Fluorescence degree in the operated rectal area and in the vaginal Suture ICG adverse reactions Operative time Digestive fistula
		2021 Raimondo [15]	Italy	Retrospective, observational, single-center, cohort, pilot study	32	May 2018–January 2020	Full-thickness bowel resection for RSE	Fluorescence degree of the anastomotic lie ICG adverse reactions Operative time Anastomotic leakage
		2021 Raimondo [14]	Italy	Retrospective, multicentric, cohort, pilot study	33	November 2019–July 2020	Full-thickness bowel resection for RSE	Accuracy of quantitative NIR–ICG evaluation in predicting bowel fistula Accuracy of qualitative NIR–ICG imaging in predicting bowel fistula Reproducibility of quantitative and qualitative NIR–ICG imaging
**Non-endometriosis**	To assess vascular perfusion of the vaginal cuff after total hysterectomy to decrease vaginal cuff dehiscence rate	2017 Beran [26]	USA	Prospective, observational cohort, single-center, pilot study	20	2 months	TLH for benign gynecologic condition	Vaginal cuff fluorescence rate Percent of cuff perimeter with adequate perfusion Length of vaginal cuff adequately perfused
		2018 Beran [27]	USA	Prospective, observational, cohort, single-center, study	20	February 2016–March 2017	RATLH for benign gynecologic condition	Vaginal cuff perfusion
	To intraoperatively evaluate ovarian perfusion after adnexal detorsion	2022 Nicholson [16]	USA	Prospective, observational, cohort, multicenter, single-arm study	12	September 2018–December 2020	Adnexal torsion	Feasibility of using ICG dye Intraoperative visualization of ICG perfusion to the detorsed adnexa Time to visualized perfusion Operative time Ovarian preservation Post operative follow-up measures

**PE**: peritoneal superficial endometriosis; **DIE**: deep infiltrating endometriosis; **ASRM**: American Society for Reproductive Medicine; **RVDIE**: rectovaginal deep infiltrating endometriosis nodules; **RSE**: rectosigmoid endometriosis; **rARSM**: revised American Society for Reproductive Medicine; **TLH**: total laparoscopic hysterectomy; **RATLH**: robot-assisted total laparoscopic hysterectomy; **-**: not reported.

**Table 2 medicina-58-00792-t002:** Patients’ characteristics.

Field	ICG Application	Study	Age, Years [Median or Mean ± SD (Range)]	BMI, kg/m^2^ [Median or Mean ± SD (Range)]	Parity	Operative time, Minutes (min) [Median or Mean ± SD (Range)]	Follow up Time (Months) [Median or Mean ± SD (Range)]	Estimated Blood Loss (mL) [Median or Mean ± SD (Range)]	Indication for Surgery
Endometriosis	To localize ureteral course	2015 Park [20]	35 ± ns	28 ± ns	1.2 ± ns	121 ± ns	5.6 ± ns	23 ± ns	Dysmenorrhea (9), dyspareunia (8), menorrhagia (7) pelvic pain (8), ovarian cysts (3), infertility (1)
		2019 Mandovra [23]	46.7 (8–78)	23.2 (21.6–32.1)	-	138 (90–240)	-	-	-
	To assess ureteral perfusion after conservative surgery	2020 Raimondo [1]	35.3 ± 6.8	24.9 ± 5.85	6 patients ≥ 1	-	-	-	Pelvic pain (16), dysmenorrhea (12), dyspareunia (16), dyschezia (10)
	To improve endometriosis identification	2018 Cosentino [7]	37 (31.5–42.5)	22 (21–24)	-	-	-	-	Dysmenorrhea (27), dyschezia (14), dysuria (5), dyspareunia (23), pelvic pain (22)
		2018 De Neef [22]	-	-	-	-	16 (2–23)	-	Symptomatic RVDIE
		2019 Jayakumaranet [12]	33 ± 2.8	28.6 ± 3	-	-	1	-	Endometriosis (3)
		2019 Lier [19]	34.5 (29.3–39.5)	<25 (12 patients- 60%) 25–30 (8 patients 40%)	0 (0–1)	30 (30–37.5 min)	-	50 (IQR: 27.5–100)	Dysmenorrhea (19), dyschezia (13), dysuria (1), dyspareunia (10)
		2020 Siegenthaler [24]	33.7 ± 6.68	23.4 ± 4.19	4 patients ≥ 1	163.5 ± ns	-	110.8 ± ns	Pelvic pain (45), infertility (4), both (13)
		2020 Vizzielli [21]	37 (31–42)	19 (19–24)	-	150 (118–185)	1	100 (50–250)	Dysmenorrhea (8), dyschezia (7), dysuria (8), dyspareunia (7), pelvic pain (7)
	To evaluate the different RSE vascular patterns and the correlation with clinicopathological data	2020 Raimondo [2]	25 ± 5.8	35.4 ± 7.2	6 patients ≥ 1	-	3	-	Dysmenorrhea (8), dyschezia (6) dyspareunia (6), pelvic pain (7), renal colic (2), hydroureter (6), hydroureteronephrosis (4)
	To assess bowel vascularization after surgery to reduce the risk of fistula	2020 Bourdel [25]	35 ± 6.7	25 (22.7–30.8)	-	240 (180–254)	3	-	-
		2021 Raimondo [15]	36 ± 7	26 ± 6.4	8 patients ≥ 1	210 (95–300)	3	125 (100–500)	Dysmenorrhea (8), dyschezia (6), dyspareunia (5), pelvic pain (7), rectorrhagia (1)
		2021 Raimondo [14]	35.1 ± 6.2	22.8 ± 5.2	5 patients ≥ 1	180 (70–350)	3	100 (10–150)	Dysmenorrhea (8), dyschezia (6), dyspareunia (6), pelvic pain (4)
Non-endometriosis	To assess vascular perfusion of the vaginal cuff after total hysterectomy to decrease vaginal cuff dehiscence rate	2017 Beran [26]	45.5 (32–68)	30.4 (22.4–44.7)	1.5 (0–4)	-	3	150 (20–450)	Pelvic pain (4), AUB (17)
		2018 Beran [27]	45 (31–64)	28.0 (21.1–43.6)	2 (0–3)	-	3	65.5 (25–400)	AUB (10), Pelvic pain (5), cervical dysplasia (3), Lynch syndrome (1), postmenopausal bleeding (1)
	To intraoperatively evaluate ovarian perfusion after adnexal detorsion	2022 Nicholson [16]	27 (25–31)	-	-	73.4 (48–94)	1	-	Suspected adnexal torsion
**TOTAL**	-	-	25–36 (mean)	22.8–35.4 (mean)	14.9% patients ≥ 1	121–163.5 (mean)	1–23 (n)	50–150 (median)	43.8% dysmenorrhea 27.6% dyschezia 14.9% dysuria 34.5% dyspareunia 39.3% pelvic pain 30.0% ovarian cysts 6.8% infertility 20.6% pelvic pain and infertility 6.7% renal colic 20.0% hydroureter 13.3% hydroureteronephrosis 3.1% rectorrhagia 42.5% AUB 15.0% cervical dysplasia 5.0% Lynch syndrome 5.0% postmenopausal bleeding 26.7% menorrhagia

**-**: not reported; **PMB**: post-menopausal bleeding; **AUB**: abnormal menstrual bleeding; **IQR**: interquartile range; ns: not stated.

**Table 3 medicina-58-00792-t003:** Details about ICG and surgery.

Field	ICG Application	Study	Surgical Procedure and Detection System of Fluorescence	Indocyanine Dosage and Injection Method	Time to ICG Visualization in Minutes [Median or Mean ± SD (Range)]	Type of Surgery
**Endometriosis**	To localize ureteral course	2015 Park [20]	Robotic-assisted laparoscopy	- intraurethral	-	Resection of deep infiltrating endometriosis, ureterolysis and bilateral ureteral stent placement and removal
		2019 Mandovra [23]	Laparoscopy	5 mg ICG diluted in 2 mL of distilled water–cystoscopy and ureteric cannulation	7 (6–9)	Ventral mesh rectopexy Rectopexy Sacrocolpopexy Anterior resection Sigmoid colectomy Right hemicolectomy Total colectomy Hysterectomy Endometriotic cyst excision
	To assess ureteral perfusion after conservative surgery	2020 Raimondo [1]	Laparoscopy	0.25 mg/kg-intravenous	5.4 ± 2.3	Removal of deep endometriotic lesions of the posterior and anterior compartments
	To improve endometriosis identification	2018 Cosentino [7]	Laparoscopy	0.25 mg/kg-intravenous	5–30	-
		2018 De Neef [22]	Laparoscopy	0.25 mg/kg-intravenous	-	Laparoscopic shaving
		2019 Jayakumara [12]	Robotic-assisted laparoscopy	0.25 mg/kg-intravenous	-	Robotic endometriosis resection
		2019 Lier [19]	Laparoscopy	Bolo of 1 mL-intravenous	5 ± ns	-
		2020 Siegenthaler [24]	Laparoscopy	0.3 mg/kg-intravenous 25 mg-intraurethral	2–20	-
		2020 Vizzielli [21]	Robotic-assisted laparoscopy and Laparoscopy	0.25 mg/kg-intravenous	15–30	Ovarian cyst removal Peritoneal removal Retrocervical nodule removal Vaginal nodule removal Utero-sacral ligament nodule removal Rectal nodule shaving Resection and anastomosis of sigma-rectum Resection and anastomosis of sigma-rectum plus loop ileostomy Discoid resection of bowel Appendicectomy Salpingectomy Ureteral stent placement, bladder surgery
	To evaluate the different RSE vascular patterns and the correlation with clinicopathological data	2020 Raimondo [2]	Robotic-assisted laparoscopy and Laparoscopy	0.25 mg/kg-intravenous	(5–50) s	RSE: Shaving, Discoid resection, Segmental resect
	To assess bowel vascularization after surgery to reduce the risk of fistula	2020 Bourdel [25]	Laparoscopic	0.2 mg/kg-intravenous	60 (45–60) s	Rectal shaving
		2021 Raimondo [15]	Laparoscopy	0.25 mg/kg-intravenous	33 (6−41) s	Discoid excision and segmental resection Hysterectomy Salpingectomy Ovariectomy Protective ileostomy
		2021 Raimondo [14]	Laparoscopy	0.25 mg/kg-intravenous	30 (9–43) s	Discoid excision and segmental resection Hysterectomy Salpingectomy
**Non-endometriosis**	To assess vascular perfusion of the vaginal cuff after total hysterectomy to decrease vaginal cuff dehiscence rate	2017 Beran [26]	Laparoscopy	25 mg + 2.5/5 mg-intravenous	11 s	Total laparoscopic hysterectomy
		2018 Beran [27]	Robotic-assisted laparoscopy	2.5–10.0 mg followed by a 10 mL saline flush-intravenous	18.4 ± 7.3 s before cuff closure 19 ± 8.7 after cuff closure	Robot-assisted total laparoscopic hysterectomy
	To intraoperatively evaluate ovarian perfusion after adnexal detorsion	2022 Nicholson [16]	Laparoscopy	8–20 cc-intravenous	1 (1–2)	Adnexal detorsion-annessectomy

**-**: not reported; **ns**: not stated; **ICG**: Indocyanine Green; **NIR**: near infra-red; **DIE**: Deep infiltrating endometriosis; **PE**: peritoneal superficial endometriosis; **RSE**: rectosig-moid endometriosis; **NPV**: negative predictive value; **PPV**: positive predictive value; **RVDIEN**: rectovaginal DIE nodules.

## Data Availability

Not applicable.

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
