# Peer review of "The Use of near Infra-Red Radiation Imaging after Injection of Indocyanine Green (NIR–ICG) during Laparoscopic Treatment of Benign Gynecologic Conditions: Towards Minimalized Surgery. A Systematic Review of Literature"

_medicina, 2022, doi:10.3390/medicina58060792_

Round 1
Reviewer 1 Report
I read with great interest the Systematic Review titled “The use of near infra-red radiation imaging after injection of indocyanine green (NIR-ICG) during
laparoscopic treatment of benign gynecologic conditions: a systematic review”, which falls whithin the aim of Medicina. The authors intend to calculate the influence of near-infrared radiation imaging following indocyanine green injection during laparoscopic treatment on benign gynecologic pathologies. The paper is well written, and the topic is interesting enough to attract the readers’ attention. Furthermore, the review methodology is accurate, and the outcomes are interesting enough to encourage further research, particularly for deep infiltrating endometriosis.
Author Response
REVIEWER #1
Comment #0
The paper is well written, and the topic is interesting enough to attract the readers’ attention. Furthermore, the review methodology is accurate, and the outcomes are interesting enough to encourage further research, particularly for deep infiltrating endometriosis.
- Authors response: We thank the Reviewer for the kind comment.
- Location: -
Reviewer 2 Report
This kind of semantic review assesses the applications of ICG in laparoscopic benign gynaecological surgery. The topic is challenging as for all sub-procedures during MIS, including endometriosis, the lack of good evidence hinders the final evaluation of this method in both diagnostic and therapeutic scenarios. The topic requites an analytical, well-designed, prospective study. Suggestions to improve the quality of the manuscript prior to publication:
- The PICO classification, albeit valid creates confusion because it assumes there is a control group. If that were the case, then a meta-analysis for different outcomes might have been possible.
- It might worth on the tables of studies to add a column reporting the outcomes of the individual studies.
- An example of a search strategy for at least one of the individual databases, for example MEDLINE should be reported.
- The take on message is not the feasibility of ICG application but that ICG might change the way we perform laparoscopic gynaecological surgery; perhaps from MIS to minimalise surgery if we want to play with the words.
- The Discussion section is exhaustive to read. Subheadings critically appraising the evidence for individual subprocedures/outcomes would facilitate reading i.e, endometriosis, DIE, shaving of an endometriosis nodule off the bladder, assessment of vault perfusion at MIS, visualization of ureters etc. Clarification is required on the role of ICG in the detection of endometriosis as for neovascularization to exist, a lesion >2cm3 is required, hence the role of ICG in the detection of endometriosis is incosistent. The SR of Ianieri M ,et al, 202 needs to be referenced as an additional effort on the subject.
- A possible role-added value to be explored is that of ICG during myomectomy. The concept is that, as we usually clamp uterine arteries, IV use of ICG may detect uterine vascularization, hence any vessels we would like to avoid during myomectomy. Any reference for that?
- The added value of detecting suspected intra-operative complications given that ICG can stay detectable for almost 6 hours.
- In the introduction, I suggest you quote one of the pioneer trials for the use of ICG in gynaecological cancers collectively (Laios A et al, 2015).
Author Response
REVIEWER #2
Comment #0
The topic is challenging as for all sub-procedures during MIS, including endometriosis, the lack of good evidence hinders the final evaluation of this method in both diagnostic and therapeutic scenarios. The topic requites an analytical, well-designed, prospective study.
- Authors response: We thank the Reviewer for the comments and we agree.
- Location: -
Comment #1
Suggestions to improve the quality of the manuscript prior to publication:
The PICO classification, albeit valid creates confusion because it assumes there is a control group. If that were the case, then a meta-analysis for different outcomes might have been possible.
- Authors response: We thank the Reviewer for the comment. We adopted the PICO classification within the a priori study protocol. In fact, methods section was planned before the beginning of the study. Unfortunately, primary studies were poorly comparable and data were not enough to be pooled. We added this lack as a limitation in the revised manuscript in order to direct future studies.
- Location: page 19; lines 399-400
Comment #2
It might worth on the tables of studies to add a column reporting the outcomes of the individual studies.
- Authors response: We thank the Reviewer for the suggestion which allowed us to improve the manuscript. We added a column reporting the outcomes of the individual studies in the revised Table 1.
- Location: Revised Table 1
Comment #3
An example of a search strategy for at least one of the individual databases, for example MEDLINE should be reported.
- Authors response: We thank the Reviewer for the suggestion. We reported an example of a search strategy adopted for the MEDLINE database in the revised manuscript.
- Location: page 3; lines 96-100
Comment #4
The take on message is not the feasibility of ICG application but that ICG might change the way we perform laparoscopic gynaecological surgery; perhaps from MIS to minimalise surgery if we want to play with the words.
- Authors response: We thank the Reviewer for the nice and brillant comment. We added the suggested message to the conclusions section of the revised manuscript.
- Location: page 19; lines 404
Comment #5
The Discussion section is exhaustive to read. Subheadings critically appraising the evidence for individual subprocedures/outcomes would facilitate reading i.e, endometriosis, DIE, shaving of an endometriosis nodule off the bladder, assessment of vault perfusion at MIS, visualization of ureters etc. Clarification is required on the role of ICG in the detection of endometriosis as for neovascularization to exist, a lesion >2cm3 is required, hence the role of ICG in the detection of endometriosis is incosistent. The SR of Ianieri M ,et al, 202 needs to be referenced as an additional effort on the subject.
- Authors response: We thank the Reviewer for the comments. We added subheadings for facilitate reading in the revised manuscript. Additionally, we clarifiied the role of ICG in the detection of endometriosis lesions lower than 2cm3 in the revised manuscript. Lastly, we referenced the 2021 systematic review by Ianieri M et al.
- Location: page 84; lines 348-354 + References list
Comment #6
6) A possible role-added value to be explored is that of ICG during myomectomy. The concept is that, as we usually clamp uterine arteries, IV use of ICG may detect uterine vascularization, hence any vessels we would like to avoid during myomectomy. Any reference for that?
- Authors response: We thank the Reviewer for the interesting comment. The idea is brilliant, and this application appears useful and potentially feasible. We updated the literature search and we did not find related references.
- Location: -
Comment #7
7) The added value of detecting suspected intra-operative complications given that ICG can stay detectable for almost 6 hours
- Authors response: We thank the Reviewer for the comment. However, unlike other types of injection, the intravenous injection of ICG has shown a maximum wash-out time of 30 min [24]. If a longer wash-out time is confirmed, this application could be very useful.
- Location: -
Comment #8
8) In the introduction, I suggest you quote one of the pioneer trials for the use of ICG in gynaecological cancers collectively (Laios A et al, 2015).
- Authors response: We thank the Reviewer for the suggestion. We quoted the suggested study in the revised manuscript.
- Location: page 2, line 69 + References list
REVIEWER #2
Comment #0
The topic is challenging as for all sub-procedures during MIS, including endometriosis, the lack of good evidence hinders the final evaluation of this method in both diagnostic and therapeutic scenarios. The topic requites an analytical, well-designed, prospective study.
- Authors response: We thank the Reviewer for the comments and we agree.
- Location: -
Comment #1
Suggestions to improve the quality of the manuscript prior to publication:
The PICO classification, albeit valid creates confusion because it assumes there is a control group. If that were the case, then a meta-analysis for different outcomes might have been possible.
- Authors response: We thank the Reviewer for the comment. We adopted the PICO classification within the a priori study protocol. In fact, methods section was planned before the beginning of the study. Unfortunately, primary studies were poorly comparable and data were not enough to be pooled. We added this lack as a limitation in the revised manuscript in order to direct future studies.
- Location: page 19; lines 399-400
Comment #2
It might worth on the tables of studies to add a column reporting the outcomes of the individual studies.
- Authors response: We thank the Reviewer for the suggestion which allowed us to improve the manuscript. We added a column reporting the outcomes of the individual studies in the revised Table 1.
- Location: Revised Table 1
Comment #3
An example of a search strategy for at least one of the individual databases, for example MEDLINE should be reported.
- Authors response: We thank the Reviewer for the suggestion. We reported an example of a search strategy adopted for the MEDLINE database in the revised manuscript.
- Location: page 3; lines 96-100
Comment #4
The take on message is not the feasibility of ICG application but that ICG might change the way we perform laparoscopic gynaecological surgery; perhaps from MIS to minimalise surgery if we want to play with the words.
- Authors response: We thank the Reviewer for the nice and brillant comment. We added the suggested message to the conclusions section of the revised manuscript.
- Location: page 19; lines 404
Comment #5
The Discussion section is exhaustive to read. Subheadings critically appraising the evidence for individual subprocedures/outcomes would facilitate reading i.e, endometriosis, DIE, shaving of an endometriosis nodule off the bladder, assessment of vault perfusion at MIS, visualization of ureters etc. Clarification is required on the role of ICG in the detection of endometriosis as for neovascularization to exist, a lesion >2cm3 is required, hence the role of ICG in the detection of endometriosis is incosistent. The SR of Ianieri M ,et al, 202 needs to be referenced as an additional effort on the subject.
- Authors response: We thank the Reviewer for the comments. We added subheadings for facilitate reading in the revised manuscript. Additionally, we clarifiied the role of ICG in the detection of endometriosis lesions lower than 2cm3 in the revised manuscript. Lastly, we referenced the 2021 systematic review by Ianieri M et al.
- Location: page 84; lines 348-354 + References list
Comment #6
6) A possible role-added value to be explored is that of ICG during myomectomy. The concept is that, as we usually clamp uterine arteries, IV use of ICG may detect uterine vascularization, hence any vessels we would like to avoid during myomectomy. Any reference for that?
- Authors response: We thank the Reviewer for the interesting comment. The idea is brilliant, and this application appears useful and potentially feasible. We updated the literature search and we did not find related references.
- Location: -
Comment #7
7) The added value of detecting suspected intra-operative complications given that ICG can stay detectable for almost 6 hours
- Authors response: We thank the Reviewer for the comment. However, unlike other types of injection, the intravenous injection of ICG has shown a maximum wash-out time of 30 min [24]. If a longer wash-out time is confirmed, this application could be very useful.
- Location: -
Comment #8
8) In the introduction, I suggest you quote one of the pioneer trials for the use of ICG in gynaecological cancers collectively (Laios A et al, 2015).
- Authors response: We thank the Reviewer for the suggestion. We quoted the suggested study in the revised manuscript.
- Location: page 2, line 69 + References list
Thank yo
REVIEWER #2
Comment #0
The topic is challenging as for all sub-procedures during MIS, including endometriosis, the lack of good evidence hinders the final evaluation of this method in both diagnostic and therapeutic scenarios. The topic requites an analytical, well-designed, prospective study.
- Authors response: We thank the Reviewer for the comments and we agree.
- Location: -
Comment #1
Suggestions to improve the quality of the manuscript prior to publication:
The PICO classification, albeit valid creates confusion because it assumes there is a control group. If that were the case, then a meta-analysis for different outcomes might have been possible.
- Authors response: We thank the Reviewer for the comment. We adopted the PICO classification within the a priori study protocol. In fact, methods section was planned before the beginning of the study. Unfortunately, primary studies were poorly comparable and data were not enough to be pooled. We added this lack as a limitation in the revised manuscript in order to direct future studies.
- Location: page 19; lines 399-400
Comment #2
It might worth on the tables of studies to add a column reporting the outcomes of the individual studies.
- Authors response: We thank the Reviewer for the suggestion which allowed us to improve the manuscript. We added a column reporting the outcomes of the individual studies in the revised Table 1.
- Location: Revised Table 1
Comment #3
An example of a search strategy for at least one of the individual databases, for example MEDLINE should be reported.
- Authors response: We thank the Reviewer for the suggestion. We reported an example of a search strategy adopted for the MEDLINE database in the revised manuscript.
- Location: page 3; lines 96-100
Comment #4
The take on message is not the feasibility of ICG application but that ICG might change the way we perform laparoscopic gynaecological surgery; perhaps from MIS to minimalise surgery if we want to play with the words.
- Authors response: We thank the Reviewer for the nice and brillant comment. We added the suggested message to the conclusions section of the revised manuscript.
- Location: page 19; lines 404
Comment #5
The Discussion section is exhaustive to read. Subheadings critically appraising the evidence for individual subprocedures/outcomes would facilitate reading i.e, endometriosis, DIE, shaving of an endometriosis nodule off the bladder, assessment of vault perfusion at MIS, visualization of ureters etc. Clarification is required on the role of ICG in the detection of endometriosis as for neovascularization to exist, a lesion >2cm3 is required, hence the role of ICG in the detection of endometriosis is incosistent. The SR of Ianieri M ,et al, 202 needs to be referenced as an additional effort on the subject.
- Authors response: We thank the Reviewer for the comments. We added subheadings for facilitate reading in the revised manuscript. Additionally, we clarifiied the role of ICG in the detection of endometriosis lesions lower than 2cm3 in the revised manuscript. Lastly, we referenced the 2021 systematic review by Ianieri M et al.
- Location: page 84; lines 348-354 + References list
Comment #6
6) A possible role-added value to be explored is that of ICG during myomectomy. The concept is that, as we usually clamp uterine arteries, IV use of ICG may detect uterine vascularization, hence any vessels we would like to avoid during myomectomy. Any reference for that?
- Authors response: We thank the Reviewer for the interesting comment. The idea is brilliant, and this application appears useful and potentially feasible. We updated the literature search and we did not find related references.
- Location: -
Comment #7
7) The added value of detecting suspected intra-operative complications given that ICG can stay detectable for almost 6 hours
- Authors response: We thank the Reviewer for the comment. However, unlike other types of injection, the intravenous injection of ICG has shown a maximum wash-out time of 30 min [24]. If a longer wash-out time is confirmed, this application could be very useful.
- Location: -
Comment #8
8) In the introduction, I suggest you quote one of the pioneer trials for the use of ICG in gynaecological cancers collectively (Laios A et al, 2015).
- Authors response: We thank the Reviewer for the suggestion. We quoted the suggested study in the revised manuscript.
- Location: page 2, line 69 + References list
Thank you and we look forward to hearing from you.
Sincerely,
Diego Raimondo (on behalf of all authors)
u and we look forward to hearing from you.
Sincerely,
Diego Raimondo (on behalf of all authors)
Thank you and we look forward to hearing from you.
Sincerely,
Diego Raimondo (on behalf of all authors)
Round 2
Reviewer 2 Report
Congratulations for the revised manuscript. In the conclusion section, I would insist on the take on message, which is that ICG application is not just feasible but -and that is the novelty of the systematic review- it can change the way we perform laparoscopic surgery and perhaps drift from MIS to minimalised surgery. I reckon this will add merit to your manuscript. I would even challenge you to change the title of your manuscript but I am easy with that. Well done!
Author Response
REVIEWER #2
Comment #1
Congratulations for the revised manuscript.
In the conclusion section, I would insist on the take on message, which is that ICG application is not just feasible but -and that is the novelty of the systematic review- it can change the way we perform laparoscopic surgery and perhaps drift from MIS to minimalised surgery. I reckon this will add merit to your manuscript. I would even challenge you to change the title of your manuscript but I am easy with that. Well done!
- Authors response: We thank the Reviewer for the kind comments and the valuable suggestions which allowed us to improve the manuscript. We modified the title and the conclusion section of the abstract and the main text of the revised manuscript according to the suggestions.
- Location: page 1, line 3-4 page 2, lines 51-53; page 20 lines 407-409.
Thank you and we look forward to hearing from you.
Sincerely,
Diego Raimondo (on behalf of all authors)